# Correlations Between Body Composition and Aerobic Fitness in Elite Female Youth Water Polo Players

**DOI:** 10.3390/sports13020051

**Published:** 2025-02-10

**Authors:** Mark Zamodics, Mate Babity, Gusztav Schay, Tamas Leel-Ossy, Agnes Bucsko-Varga, Panka Kulcsar, Regina Benko, Dora Boroncsok, Alexandra Fabian, Adrienn Ujvari, Zsuzsanna Ladanyi, Dorottya Balla, Hajnalka Vago, Attila Kovacs, Eva Hosszu, Szilvia Meszaros, Csaba Horvath, Bela Merkely, Orsolya Kiss

**Affiliations:** 1Heart and Vascular Center, Faculty of Medicine, Semmelweis University, 1122 Budapest, Hungary; babity.mate@semmelweis.hu (M.B.);; 2Department of Sports Medicine, Faculty of Medicine, Semmelweis University, 1122 Budapest, Hungary; 3Department of Biophysics and Radiation Biology, Semmelweis University, 1094 Budapest, Hungary; 4Department of Internal Medicine and Oncology, Faculty of Medicine, Semmelweis University, 1083 Budapest, Hungary; 5Pediatric Center, Tűzoltó Street Department, Semmelweis University, 1094 Budapest, Hungary

**Keywords:** CPET, DEXA, VO2, water polo, lean body mass, body fat mass

## Abstract

Body composition and cardiopulmonary exercise testing (CPET) are vital for optimizing sports performance, but the correlations between them are still underexplored. Our study aimed to investigate the relationships between body composition and specific CPET variables describing physical fitness in young athletes, also adjusting for age and height, in a less-studied, female population. Seventy players participated in our study (age: 16.10 ± 1.63 y). After determining body composition using dual-energy X-ray absorptiometry, we conducted treadmill-based maximal-intensity CPET. Data were analyzed in R using multivariate linear regression, accounting for age and height as confounders. Lean body mass (LBM), body fat mass (BFM), and bone mineral content (BMC) showed no effect on resting, maximum, or recovery heart rates and no correlation with resting or maximal lactate values. LBM positively correlated with maximum ventilation (VE-max) (Est: 1.3 × 10^−3^; SE: 6.1 × 10^−4^; *p* < 0.05) and maximum absolute oxygen consumption (VO2_abs_-max) (Est: 7.710^−5^; SE: 6.9 × 10^−6^; *p* < 0.001)—with age as an influencing factor for VE-max and height as an influencing factor for VO2_abs_-max. Conversely, BFM showed a negative correlation with maximum relative oxygen consumption (VO2_rel_-max) (Est: −4.8 × 10^−4^; SE: 1.2 × 10^−4^; *p* < 0.001). Moreover, BFM and BMC were also negatively correlated with maximal exercise duration (Est: −2.2 × 10^−4^; SE: 8.0 × 10^−5^; *p* < 0.01; Est: −3.2 × 10^−3^; SE: 1.4 × 10^−3^; *p* < 0.05) with height as an influencing factor. Our findings indicate complex correlations between body composition and CPET parameters, providing important information for the analysis of individual ergospirometric data. Our results draw attention to the fact that body composition is more precise than weight and height in the evaluation of athletes’ physical fitness.

## 1. Introduction

Cardiopulmonary exercise testing (CPET) is widely utilized in both cardiology and pulmonology practices [1,2]. It also provides clear insight into the performance of the cardiovascular, pulmonary, and metabolic systems. It is highly suitable for monitoring the physical capacity of patients, as well as investigating stress-related symptoms and assessing pre- and post-disease and operative conditions [3,4]. Furthermore, assessments of maximal-intensity CPET results offer useful information for diseased athletes [5]. At present, sports cardiology screenings of healthy athletes are imperative to a safe and effective sports performance. CPET examination is an integral part of this extended screening, aiding in the identification of heart and lung diseases as well as presenting the scope of cardiorespiratory fitness measurements [6,7]. Regular CPET measurements also serve to monitor the progress of an athlete and assist in developing training plans. Various commonly used CPET parameters, such as maximal oxygen uptake, ventilation, resting and maximal heart rate, and lactate values, are indicative of the actual fitness level of an athlete [8].

Alongside respiratory and cardiac parameters, changes in body composition as a result of adaptation to sport also play a significant role in determining physical fitness [9]. By determining muscle mass, fat mass, fat-free mass, body fat percentage, and bone mineral content as resting determinants of physical fitness, changes in the body composition of an athlete due to training, diet, or growth can be assessed [10,11]. Although a number of methods are used, dual-energy X-ray absorptiometry (DEXA) is a highly reliable and widely available method for determining body composition [12]. DEXA uses two different energy levels of X-ray photons passing through the body, allowing for the determination of a three-component model that includes body fat mass (BFM), lean body mass (LBM), and bone mineral content (BMC). Furthermore, DEXA is capable of evaluating the amounts of android and gynoid fat, as well as the android-to-gynoid fat ratio, which has significant prognostic value [13].

Previous research has shown a strong correlation between body composition and muscle strength, as well as between body composition and cardiac muscle mass, in athletes [14,15,16]. These research findings suggest the need for a broader application and investigation of body composition parameters in the screening and follow-up of athletes. Youth athletes, as a vulnerable population due to the intensive training load that occurs during biological growth, require special attention in terms of monitoring proper body composition and aerobic fitness development. Test results for male and female athletes differ in a number of aspects and there are some specific aspects that must be considered in examinations of female athletes (e.g., the female athlete triad) [17]. However, youth athletes and females are generally underrepresented in scientific reports on elite athletes [18]. Most of the experimental and clinical studies regarding elite athletes examine adult male populations, as more information is available, and more male athletes are clinically available to obtain scientific measurements. However, there is a need for a detailed examination of, and research on, youth and female athletes as well. Understanding the responses of these specific populations to exercise compared to those of adults or males can lead to a safer and more effective approach to youth and female sports. Studies like ours could encourage more and more young female athletes to engage in physical activity and adopt a healthy lifestyle.

In Hungary, water polo is one of the most scientifically researched and popular sports, with athletes achieving excellent international results at both adult and youth levels [19,20]. Due to the mixed nature of water polo, it develops both endurance and muscle strength, and regular sports cardiology examinations, including CPET and body composition analysis, contribute greatly to the successful sports performance of elite water polo players. Although CPET and body composition assessments are increasingly used in sports, there is limited research available on their combined evaluation. In the present study, we aimed to perform a correlational analysis of body composition parameters and certain CPET variables describing physical fitness in a special population of young elite water polo players.

## 2. Material and Methods

### 2.1. Study Population

A total of 70 asymptomatic (without complaints and without signs of diseases or injuries), elite (members of national teams, Olympians, and professional athletes, who generally exercise >10 h/week), female youth (<20 years old) water polo players, members of national youth teams, participated in our comprehensive cardiology screening program at the Heart and Vascular Center of Semmelweis University, with an age range of 13.8–19.9 years and an average age of 16.10 ± 1.63 years. The age groups of U16 (age < 16, N = 39), U18 (age 16–18, N = 19), and U20 (age 18–20, N = 12) competition classes were also analyzed [21,22]. The average training time was 16.80 ± 5.28 h per week. Among the players, there were 10 goalkeepers and 60 field players (36 wings–flats, 10 centers, and 14 defenders).

All athletes included in the present study were free from injuries or health conditions (with particular regard to significant cardiovascular diseases, infections, metabolic diseases, and psychological issues) that could have affected the results. Following the cardiology screening detailed in Section 2.2, athletes with positive findings that could influence the results were not included in this study. All examinations were carried out in the competition season.

Before the commencement of the study, written informed consent was obtained from all participants, and from their parents and/or legal guardians in the case of athletes under 18 years of age. The informed consent sheet contained all information about the planned examinations, as well as including information on the small amount of ionizing radiation administered during the DEXA measurements. Furthermore, we specifically emphasized that the examination is contraindicated during pregnancy. The study was approved by the Medical Research Council of Hungary (No. IV/10282-1/2020/EKU) in accordance with the ethical guidelines of the Helsinki Declaration and Good Clinical Practice. All participants were in good health and participated in regular maximal-intensity training prior to the tests. All measurements were conducted at least 12 h after the last training session or match.

### 2.2. Cardiology Screening

As part of the extensive cardiology screening, each athlete completed a detailed medical history questionnaire and underwent a physical examination, body composition analysis, resting 12-lead electrocardiogram (ECG), blood pressure measurement, laboratory tests, echocardiography, and cardiopulmonary exercise testing. Athletes for whom pathological findings were obtained that could influence the results of the study were excluded. All examinations were performed under the supervision of a specialist in cardiology and sports medicine.

### 2.3. Body Composition Analysis

We used the Prodigy DEXA (GE-Lunar Corp., Madison, WI, USA) to determine body composition, specifically BMC, LBM, BFM, and the android/gynoid (A/G) fat ratio. Players wore light clothing, with all metal and plastic artifacts removed. Standing height was measured with a stadiometer to the nearest 0.1 cm and recorded as the mean of two consecutive measurements (BSM 370, InBody Co., Ltd., Seoul, Republic of Korea). Body weight was measured with a calibrated scale with a precision of 0.1 kg (BSM 370, InBody Co., Ltd., Seoul, Republic of Korea).

It is important to highlight that while DEXA utilizes ionizing radiation, the exposure level using the current models of DEXA scanners is very low (effective doses: 2–10 μSv, roughly equivalent to the dose received from natural background radiation in a day) [23]. Expert reports confirm the safe application of DEXA in sports performance assessments as part of recognized training programs for body composition analyses if carried out by properly accredited individuals. However, they emphasize that taking multiple measurements within a short period can lead to cumulative exposure; therefore, this is not recommended [23]. Our athletes were measured only once, and they were informed about the ionizing radiation that was administered (see Section 2.1).

### 2.4. Cardiopulmonary Exercise Testing

During the CPET examination, athletes performed a specific running protocol on a treadmill ergometer (T-2100, GE Healthcare, Helsinki, Finland) to reach their maximal physiological load. The protocol started with a 1 min resting sitting measurement, followed by 2 min of 6 km/h warm-up walking. The speed of the treadmill was then increased to 8 km/h and this speed was maintained while the treadmill incline increased by 1.5% per minute until the end of the maximal-intensity exercise session. This was followed by a 1 min cool-down walking period and 4 min of rest. The test was terminated after 5 min of recovery after blood pressure and lactate were measured. Respiratory volume and gas volume (oxygen uptake and carbon dioxide production) were continuously monitored (Respiratory Ergostik, Geratherm, Geratal, Germany). A continuous 12-lead electrocardiogram (ECG) (CAM-14 module, GE Healthcare, Finland) was recorded throughout the entire test. Capillary blood lactate levels were determined through fingertip sampling at rest, every second minute during exercise, at maximal load, and in the fifth minute of the recovery phase (Laktate Scout 4+, EKF Diagnostik, Barleben, Germany). Regular blood pressure measurements were carried out at rest, every third minute during exercise, at maximal load, and at the first and fifth minutes of the recovery phase.

### 2.5. Correlational Analysis

Correlations between body composition parameters—BMC, LBM, BFM, android/gynoid (A/G) fat ratio—and CPET variables—maximal exercise ventilation (VE-max), maximum absolute oxygen consumption (VO2_abs_-max), maximum relative oxygen consumption (VO2_rel_-max), maximal CPET exercise duration (Time), resting heart rate (HR), maximum HR achieved, recovery 1 min HR, 1 min pulse recovery (maximum HR−recovery 1st minute HR), resting, maximal, and 5 min recovery lactate values—were investigated, with age and height as potential confounders.

### 2.6. Statistical Analysis

Data were analyzed using R statistical software (R Core Team, version: 4.2., 2017. R: A Language and Environment for Statistical Computing. R Foundation for Statistical Computing, Vienna, Austria. https://www.R-project.org/) with multivariate linear regression models. In addition to the main predictor variable, the models included two potential confounders in each analysis: age and height. We also performed a residual analysis, collinearity assessment, confidence interval determination, and power analysis. During the collinearity assessment, the Variance Inflation Factor (VIF) values were below 2.5 in all cases. The results are provided in the Appendix A. The overall correlation coefficient was adjusted to account for the number of ordinal and numeric variables in the model. Correlations were described as estimate (Est), representing the estimated change in Y for a one-unit change in X, along with the standard error (StE). Statistical significance was defined as *p* < 0.05. Descriptive data are presented as mean ± SD.

## 3. Results

### 3.1. Maximal Exercise Ventilation (VE-Max)

LBM showed a positive correlation with VE-max [L/min], whereas age had a negative effect (multiple R^2^: 0.16; adjusted R^2^: 0.12) (Table 1). Other body composition parameters that were examined, such as BFM, BMC, and weight, showed no correlation with VE-max.

### 3.2. Maximum Absolute Oxygen Consumption (VO2_abs_-Max)

There was a positive correlation between LBM and VO2_abs_-max, whereas height had a negative effect and age had no significant role (multiple R^2^: 0.68; adjusted R^2^: 0.67) (Table 2, Figure 1). Furthermore, we found a positive correlation between VO2_abs_-max and BFM (multiple R^2^: 0.24; adjusted R^2^: 0.21), BMC (multiple R^2^: 0.22; adjusted R^2^: 0.18), weight (multiple R^2^: 0.48; adjusted R^2^: 0.46), although these correlations were relatively weaker (see data in the Appendix A).

### 3.3. Maximum Relative Oxygen Consumption (VO2_rel_-Max)

The strongest correlation was observed for BFM, which had a negative effect on VO2_rel_-max [mL/kg/min] without any influencing effect of age and height (multiple R^2^: 0.26, adjusted R^2^:0.23) (Table 3). Weight also negatively affected the VO2_rel_-max without the impact of confounders, though this correlation was weaker than that observed with BFM (multiple R^2^: 0.16, adjusted R^2^: 0.13). No correlation was found between LBM and VO2_rel_-max, nor between BMC and VO2_rel_-max. The A/G fat ratio negatively correlated with the VO2rel-max, whereas height acted as a negative confounding factor (multiple R^2^: 0.16, adjusted R^2^: 0.13) (see data in the Appendix A).

### 3.4. Maximal CPET Exercise Duration (Time)

BFM showed a negative correlation with exercise duration, whereas height played a negative influencing role (multiple R^2^: 0.30, adjusted R^2^: 0.27) (Table 4.). Similarly to BFM, BMC also had a negative impact on time and height acted as a confounding factor (multiple R^2^: 0.27, adjusted R^2^: 0.24). The effect of A/G fat ratio and weight approached significance (*p* = 0.079 and *p* = 0.088, respectively) in association with time, with an overall negative trend observed (see data in the Appendix A).

### 3.5. Heart Rate (HR)

No correlation was found with either the body weight or body composition parameters for resting, maximum HR achieved, recovery 1 min HR, and 1 min pulse recovery (maximum HR–recovery 1st minute HR). Furthermore, age was a consistent negative confounder for maximum heart rate (see data in the Appendix A).

### 3.6. Lactate Levels

Similarly to the heart rate parameters, we found no correlation between resting, maximal, or 5 min recovery lactate values and body weight or body composition parameters (see data in the Appendix A).

### 3.7. Our Results Based on Age Group (U16, U18, U20)

We also analyzed our key findings based on age group (U16, U18, U20). No correlation was found between LBM and VE-max in any of the age groups. However, a strong correlation was observed between LBM and VO2_abs_-max across all age groups. BFM showed a negative correlation with VO2_rel_-max in the U16 and U18 groups, while no correlation was identified in the U20 group. Additionally, BFM exhibited a negative correlation with time in the U18 and U20 groups. In contrast, height negatively influenced exercise duration, while BFM had no observed effect on this among the U16 group (see data in the Appendix A).

## 4. Discussion

Our findings suggest that body composition parameters are correlated with ergospirometry parameters. These results clearly show that resting and exercise physical fitness parameters are interrelated and cannot be analyzed separately.

We found a positive correlation between LBM and VE-max, but age had a negative effect. Although the effect of height on VE-max was not significant in our study, this is probably because LBM is closely related to height and already includes its effects. The negative effect of age on the correlation between LBM and VE-max means that a younger athlete with a similar LBM (indicating more intensive trainings at a younger age) will have a higher VE-max than an older athlete with the same LBM in our female youth population. It is also likely that if we were to extend our research to male water polo players, sex would also have an influential role. In a previous study, Kaminsky and colleagues estimated VE-max based on sex, age and height [24]. Although their study did not examine the role of body composition and the population was not athletic, their findings were consistent with ours and male sex was also associated with higher VE-max values based on their results. D. A. Quesada and colleagues also demonstrated the importance and predictive role of VE-max in healthy individuals in their research, highlighting the need to establish appropriate reference values [25]. Overall, it can be concluded that utilizing the relationship between body composition and VE-max provides further opportunities to establish VE-max reference ranges for elite athletes.

Similarly to VE-max, we also found a correlation between VO2_abs_-max and certain body composition parameters in youth female elite athletes. LBM emerged as the most influential body composition parameter for VO2_abs_-max, alongside the influencing effects of height. The negative effect of height on this correlation indicates that, among youth female athletes with equal LBMs, the shorter individual will have a higher VO2_abs_-max, as they possess a greater muscle mass relative to their height. It is important to emphasize that our study simultaneously examined the effects of LBM, height, and age. Among this triad, LBM emerged as the primary determinant of oxygen uptake capacity. Since height positively influences LBM to a certain extent, its otherwise expected positive impact diminishes. Furthermore, in our equation, height is now represented as a negative influencing factor. Although previous studies have described positive correlations between height and VO2, they did not account for the role of LBM [26,27].

We also found a positive, albeit relatively weaker correlation between VO2_abs_-max and BFM, BMC, and weight. With these correlations, resting fitness parameters, such as LBM, in combination with anthropometric data, can be used to estimate the VO2_abs_-max and help establish normal reference values. Previous studies have underscored the significance and potential of these associations; however, sport-specific normal CPET values are lacking in the literature [28,29]. In line with our results, C.H. Kim and colleagues confirmed a strong correlation between muscle mass and VO2abs-max in their study of kayakers on two different ergometers. Their research also attributed significant importance to age, probably due to the older (50+) population they studied [30].

On the basis of our research findings and in contrast to the VO2_abs_-max, the VO2_rel_-max is primarily influenced by fat values. BFM showed a negative correlation with VO2_rel_-max without significant confounders. As regular aerobic exercise decreases BFM and increases VO2_rel_-max, this correlation is not surprising. The negative relationship between BFM and VO2_rel_-max has also been documented in previous studies [29,31]. Although LBM showed a positive correlation with VO2abs-max, we could not find the same correlation for VO2rel-max. This is probably due to the dimension of relative VO2 (mL/kg/min), as the weight distribution may mask the effect of LBM on relative maximal aerobic capacity.

In line with the negative effect of increasing BFM on VO2rel-max, we also found a negative correlation between exercise time and BFM. Again, this result is not surprising, as BFM is not actively involved in movement and therefore only acts as excess weight, making physical exercise more difficult [29,31]. Similarly to BFM, BMC also showed a negative correlation with exercise duration, probably for similar reasons as those mentioned above: in some respects, BMC is also an excess weight that needs to be carried by the working muscles as the active part of the skeletomuscular system during running. In both cases (BFM and BMC), height had a negative effect, indicating that considering anthropometric data is also essential in this context. The impact of height is particularly notable, as athletes with the same BMC tend to exercise for longer durations if they are shorter. This is not surprising, since shorter athletes with the same BMC have a higher bone density due to their more intense physical activity [32,33]. It is important to emphasize that the role of the skeletal system is essential to establish the optimal sport adaptation processes of elite athletes.

The associations between exercise time and weight, as well as between exercise time and the A/G fat ratio, were found to be almost significant when height was used as a negative confounder. It is hypothesized that increasing the sample size would reveal a significant relationship between weight and time, and also between the A/G fat ratio and time.

We found a negative correlation between the A/G ratio and VO2_rel_-max, with height also having a negative influencing effect. Although this relationship was previously established in patients with heart failure, it has not been documented in elite athletes [34]. These results are in line with the literature data on the negative effect of android-type fat mass gain on cardiovascular morbidity and mortality.

J. Lach and colleagues observed correlations between body composition and maximal heart rate. However, the effects of these parameters were overshadowed by the effect of age; therefore, their use is less recommended [35]. Previous studies have described associations between body composition and resting heart rate and heart rate recovery; however, these investigations did not focus on elite athletes [36,37,38]. In contrast to these findings, our study identified no significant correlations between resting, maximal, or recovery heart rates and body composition parameters. This discrepancy is probably due to our special population of female youth elite athletes. Although, as in previous studies, we found no correlation with body composition, we did observe a strong relationship between age and maximum heart rate, which still appears to be the best predictor of maximum heart rate [39,40]. The results are likely attributable to two main factors. First, water polo is a mixed skill-based sport, where athletes may exhibit varying levels of aerobic and anaerobic fitness. Moreover, some players may have been selected primarily for their tactical and technical skills rather than their endurance capabilities. The second factor is the elite status of the athletes; differences in body composition are less pronounced compared to those of non-athletes or amateur athletes. Consequently, in this group, age becomes the most significant determining factor.

No correlation was found between resting and exercise lactate levels and body composition parameters. The reason for this somewhat surprising finding is presumably due to the specific sport activity of our athlete group. Water polo is a mixed sport that includes both static and dynamic elements, requiring both good aerobic and anaerobic endurance [41]. Similarly, to heart rate, the lack of correlation is likely due to the fact that some clubs may prioritize endurance training, while others may emphasize tactics and ball skills, which demand greater anaerobic exercise tolerance and higher lactate levels. Additionally, the varying positions within a water polo team require different levels of aerobic and anaerobic fitness, as well as different levels of lactate tolerance.

Okano and colleagues found correlations between body fat ratio, body water content, and maximal lactate in healthy, non-athlete male individuals [42]. Additionally, Yui and their team discovered a positive correlation between lactate reduction and skeletal muscle mass in their study on a non-athlete population [43]. The discrepancies between the aforementioned studies and our current research may result from differences in the populations studied and the specific characteristics of water polo, as detailed above. However, these results certainly need further investigation.

Regarding our analysis of age groups (U16, U18, and U20), no correlation was found between LBM and VE-max within the groups, in contrast to the positive correlation observed in the total population. This discrepancy can be attributed to the reduced sample size, which consequently leads to an increase in standard error in subgroup analyses. A positive correlation between LBM and VO2_abs_-max was observed in all age groups, highlighting the robustness of this relationship, as this correlation persisted even with smaller sample sizes. BFM showed a negative correlation with VO2_rel_-max in the U16 and U18 age groups, while no correlation was observed in the U20 group. Similarly, BFM and time demonstrated a negative correlation in the U18 and U20 groups, but showed no correlation in the U16 group. These findings are likely attributable to the limited sample size and increased standard error. The uncertainty in the U16 results may also stem from the fewer years spent training compared to the other groups, meaning that, in the case of this group, factors other than training (e.g., genetics) may play a larger role. In the future, we plan to increase the sample size of all age groups, which will facilitate a more detailed analysis of youth groups.

## 5. Limitations

It is crucial to note that our study was based on measurements of young female athletes between the ages of 13 and 20 years, who are still developing and have neither reached their peak bone density and mass nor their peak muscle mass [44,45]. However, as scientific data on the sport adaptation processes of youth athletes are limited, this is also a major strength of our examinations.

Due to difficulties in fitting the complex tests into a busy training schedule, we did not align the examinations with the menstrual cycles of the athletes. Given that water polo athletes compete every weekend, they should be able to perform at their maximum regardless of their menstrual cycle. There are conflicting data in the literature regarding the relationship between the phases of the menstrual cycle and exercise performance. Most studies found either minimal or no association between the menstrual cycle and exercise performance [46,47,48]. Future research and personalized evaluations are necessary.

The results obtained cannot be directly applied to elite athletes in other sports or different age groups, nor to the entire population. However, our results may be useful for screening members of similar mixed sports, as no sport- and age-specific recommendations have been made available in the literature so far. Although this study only investigated female athletes, this can also be considered a strength, as research on female athletes is underrepresented in the literature.

## 6. Conclusions

Our research confirmed the strong correlation between body composition and maximal-intensity treadmill CPET aerobic fitness parameters in youth female water polo players. The obtained correlations are even stronger than those between body weight, height, and aerobic fitness parameters. By utilizing these results and expanding on the research, our study may help establish sport-specific normal CPET values for athletes, especially youth athletes. Based on our results, it is possible to create normative CPET values not using body weight, but rather body composition, height, and age, providing a more personalized evaluation of the youth athlete population. In the future, a specific calculator developed for the complex evaluation of body composition and CPET data of youth athletes—possibly through providing percentile values—would facilitate the practical application of our results. These more sensitive results could assist both the athlete and the coach in optimizing training plans and assessing individual deficiencies and help physicians in making a more accurate diagnosis, which may contribute to safer and more effective athletic development and performance.

## Figures and Tables

**Figure 1 sports-13-00051-f001:**
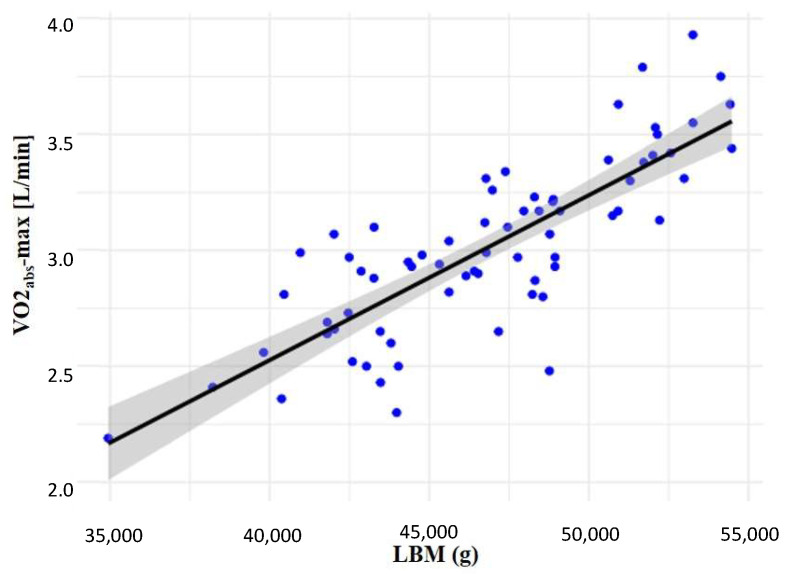
Correlation between LBM and VO2_abs_-max. Abbreviations: LBM: lean body mass; VO2_abs_-max: maximal absolute oxygen consumption.

**Table 1 sports-13-00051-t001:** Correlation between LBM and VE-max.

	Estimate	Std. Error	t-Value	*p*
Intercept	2.01 × 10^2^	7.39 × 10^1^	2.78	<0.01
LBM (g)	1.32 × 10^−3^	6.07 × 10^−4^	2.17	<0.05
Height (cm)	−3.7 × 10^−1^	4.54 × 10^−1^	−8.1 × 10^−1^	0.42
Age (years)	−4.54	1.45	−3.12	<0.01

Abbreviations: LBM: lean body mass; Std. Error: standard error; VE-max: maximal ventilation.

**Table 2 sports-13-00051-t002:** Correlation between LBM and VO2_abs_-max.

	Estimate	Std. Error	t-Value	*p*
Intercept	1.33	8.36 × 10^−1^	1.59	0.12
LBM (g)	7.69 × 10^−5^	6.85 × 10^−6^	1.12 × 10^1^	<0.001
Height (cm)	−1.18 × 10^−2^	5.14 × 10^−3^	−2.3	<0.05
Age (years)	7.75 × 10^−3^	1.64 × 10^−2^	4.71 × 10^−1^	0.64

Abbreviations: LBM: lean body mass; Std. Error: standard error; VO2_abs_-max: maximal absolute oxygen consumption.

**Table 3 sports-13-00051-t003:** Correlation between BFM and VO2_rel_-max.

	Estimate	Std. Error	t-Value	*p*
Intercept	6.94 × 10^1^	1.60 × 10^1^	4.34	<0.001
BFM (g)	−4.76 × 10^−4^	1.18 × 10^−4^	−4.05	<0.001
Height (cm)	−8.9 × 10^−2^	9.52 × 10^−2^	−9.3 × 10^−1^	0.35
Age (years)	1.74 × 10^−2^	3.03 × 10^−1^	5.8 × 10^−2^	0.95

Abbreviations: BFM: body fat mass, VO2_rel_-max.: maximal relative oxygen consumption.

**Table 4 sports-13-00051-t004:** Correlation between BFM and time.

	Estimate	Std. Error	t-Value	*p*
Intercept	4.96 × 10^1^	1.09 × 10^1^	4.53	<0.001
BFM (g)	−2.24 × 10^−4^	8.02 × 10^−5^	−2.79	<0.001
Height (cm)	−1.98 × 10^−1^	6.5 × 10^−2^	−3.05	<0.001
Age (years)	2.04 × 10^−1^	2.07 × 10^−1^	9.9 × 10^−1^	0.33

Abbreviations: BFM: body fat mass, Time: maximal CPET exercise duration.

## Data Availability

The datasets generated and analyzed in the current study are not available publicly, but are available upon reasonable request from the corresponding author.

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
