# Peer review of "Correlations Between Body Composition and Aerobic Fitness in Elite Female Youth Water Polo Players"

_sports, 2025, doi:10.3390/sports13020051_

Round 1

Reviewer 1 Report

Comments and Suggestions for Authors

Dear Authors,

Firstly, I wish you a Happy New Year. Concerning Your submission, please check my appreciation below.

The article "Correlations Between Body Composition and Aerobic Fitness in Elite Youth Water Polo Players," explores the relationship between body composition parameters and aerobic fitness outcomes in a specific population of young female athletes. The study employs robust methodologies, including dual-energy X-ray absorptiometry (DEXA) and cardiopulmonary exercise testing (CPET), to provide valuable insights into the physiological factors influencing athletic performance. The manuscript is well-structured and written in clear English, making it accessible to a broad audience. However, some aspects could be improved to enhance the clarity, methodological consistency, and practical applicability of the results. The following suggestions and indications are intended to assist the authors in strengthening the impact and quality of the manuscript.

Affiliations

- The formatting of the affiliations is not correct and needs to be revised.

Abstract

- The abstract is clear and well-structured, but it could be more specific regarding the study's objectives and key findings. Please consider rephrasing the objective to make it more direct and specific, such as: "to investigate the relationships between body composition and aerobic fitness in young athletes, adjusting for age and height."

- The objective is currently formulated in general terms and lacks emphasis on the unique aspects or practical relevance of the study. It should clarify why investigating young female athletes is important and how the findings contribute to the literature or practice.

- Some important findings, such as the absence of correlations between body composition and heart rate or lactate, are not mentioned in the abstract. Including these would provide a more comprehensive overview of the results. Please consider improving it.

- Some keywords are overly generic (e.g., "Exercise," "Ventilation") and must be more specific to increase the article visibility.

Materials and Methods

- In line 88, the phrase A total of 70 asymptomatic elite youth (<20 years old) female water polo players () is used. What is the meaning of "asymptomatic"?

- In this same paragraph, authors could expand on the sample characteristics (e.g., competitive level, positions within the team). This information, described later in the results (lines 146-148), should be included here.

- Is there a risk of bias due to the small sample size? Did the authors consider performing a power analysis to validate the results?

- The article mentions that male participants could be an influencing factor but excludes them. If the focus is exclusively on females, this should be more clearly justified, with references supporting why this population deserves special attention.

- In lines 9192, the authors state that participants were free from diseases that could influence the study. What diseases are you referring to? Please explain it carefully.

- Did the lack of consideration for the menstrual cycle potentially influence the results? Is there literature suggesting significant differences in CPET tests based on the menstrual cycle?

- In the statistical analysis subsection:

o How was potential multicollinearity between variables (e.g., height and lean body mass) addressed? Was the independence of predictors tested?

o How was the robustness of the statistical models determined? Please consider including more validation metrics, such as residual analysis or confidence intervals.

Results

- Some results seem unexpected (e.g., the absence of correlation between body composition and heart rate or lactate). Do the authors have explanations or hypotheses for these discrepancies compared to existing literature?

- The results indicate that height harms some correlations (e.g., VO2abs-max). This effect should be explored further in the discussion, with clear hypotheses explaining how height influences the findings.

- Height is mentioned as a confounding factor, but the authors do not sufficiently explain why the effect is negative or how this compares with the existing literature.

- The emphasis on correlations should not use red font but bold text for better clarity.

- The text preceding the tables is highly repetitive and should be simplified, leaving detailed explanations for the discussion section.

- In the result tables, there is inconsistency in the presentation of results, both in formatting (some are in scientific notation) and in the number of decimal places. Formatting should be consistent and appropriate for the parameters presented.

- Figure captions should be placed below the respective figures.

Conclusion

- The introduction highlights the need to establish normative values for young athletes, but the conclusion does not sufficiently develop how the findings could contribute to this.

- There should be a stronger connection between the objectives presented in the introduction and the conclusions.

- What practical impact do you expect these findings to have on training or evaluating elite athletes?

- The study limitations should be presented immediately following the discussion.

Author Response

Dear Reviewer,

Thank you very much for taking the time to review our manuscript. Your insights and comments have greatly contributed to enhancing the professionalism and value of our work. We sincerely appreciate your effort and dedication.

Best regards,
Mark Zamodics 

Affiliations

- The formatting of the affiliations is not correct and needs to be revised.

Thank you for the comment, we have made the corrections (lines 7-16).

Abstract

- The abstract is clear and well-structured, but it could be more specific regarding the study's objectives and key findings. Please consider rephrasing the objective to make it more direct and specific, such as: "to investigate the relationships between body composition and aerobic fitness in young athletes, adjusting for age and height."

Thank you for the comment, we agree. The suggested modifications have been made (lines 22-24, 39-41).

- The objective is currently formulated in general terms and lacks emphasis on the unique aspects or practical relevance of the study. It should clarify why investigating young female athletes is important and how the findings contribute to the literature or practice.

Thank you for your feedback. We have also supplemented the abstract (line 24), however, due to the character limit, we elaborated on the significance of research on female's sports in the introduction. (line 79-86).

- Some important findings, such as the absence of correlations between body composition and heart rate or lactate, are not mentioned in the abstract. Including these would provide a more comprehensive overview of the results. Please consider improving it.

Thank you, the absence of correlation between body composition and heart rate or lactate is mentioned in the abstract, although with slightly different wording (lines 29-30). Due to the character limit, it is difficult to make significant, detailed changes in the abstract. However, the absence of the upper mentioned correlations and the potential explanations for these findings are detailed and has been complemented in the Discussion (lines 294-319).   

- Some keywords are overly generic (e.g., "Exercise," "Ventilation") and must be more specific to increase the article visibility.

Thank you for the comment, we agree. We have removed "exercise" and "ventilation" and replaced them with Water polo and Body fat mass.

Materials and Methods

- In line 88, the phrase “A total of 70 asymptomatic elite youth (<20 years old) female water polo players (…)” is used. What is the meaning of "asymptomatic"?

Thank you for the comment, we have clarified the expression (lines 99, 108-111).

- In this same paragraph, authors could expand on the sample characteristics (e.g., competitive level, positions within the team). This information, described later in the results (lines 146-148), should be included here.

We have relocated lines 146-148 and further expanded the description (lines 103-106).

- Is there a risk of bias due to the small sample size? Did the authors consider performing a power analysis to validate the results?

Thank you for the question. In our study, we examined the entire Hungarian national youth female water polo team, making this the maximum sample size achievable under the circumstances. We performed a power analysis, which in all cases demonstrated a power >0.8 (see Supplementary Materials).

In the future, we plan to continue collaborating with the Water Polo Federation to examine more athletes as members of the future youth teams and we plan to conduct prospective validation studies with an independent youth female athlete group.

- The article mentions that male participants could be an influencing factor but excludes them. If the focus is exclusively on females, this should be more clearly justified, with references supporting why this population deserves special attention.

Thank you for the comment. We have updated the article accordingly and also included a reference (lines 78-86). https://doi.org/10.1177/03635465221131281. Naturally, the examination of a youth male athlete population is of importance, and we have also started to carry out the examinations in males.

- In lines 9192, the authors state that participants were free from diseases that could influence the study. What diseases are you referring to? Please explain it carefully.

Thank you for the feedback. "health conditions that can influence the exercise test results" is a broad term, and it would be challenging to specify each one, however, to enhance clarity, we identified the major categories (line 102-103). Moreover, we referred to the detailed cardiology screening of the athletes which also helped us to exclude athletes with pathologies that could influence the results (lines 109-112).

- Did the lack of consideration for the menstrual cycle potentially influence the results? Is there literature suggesting significant differences in CPET tests based on the menstrual cycle?

Thank you for the comment. This is a very challenging question. The international literature shows conflicting results on this matter; most studies suggest there are no differences across the phases of the menstrual cycle, or only minimal differences. Clearly, further research and individual consideration are needed. Accordingly, we have updated the article (lines 340-346).

Ref:

https://doi.org/10.1080/02701367.2023.2291473

https://doi.org/10.1249/MSS.0000000000003447

https://doi.org/10.1007/s40279-020-01319-3

- How was potential multicollinearity between variables (e.g., height and lean body mass) addressed? Was the independence of predictors tested?

Thank you for the comment. We have carried out tests for potential multicollinearity between predictors (e.g., height and lean body mass) using Variance Inflation Factor (VIF). All VIF values were below 2.5, indicating no significant multicollinearity. Therefore, we are confident that the predictors included in our model are sufficiently independent.

- How was the robustness of the statistical models determined? Please consider including more validation metrics, such as residual analysis or confidence intervals.

Thank you for the question! We conducted a residual analysis by plotting Residuals vs. Fitted Values and performed the Shapiro-Wilk test, which indicated that the residuals follow a normal distribution. Additionally, we carried out a confidence interval analysis and assessed multicollinearity. The data have been supplemented with additional information provided in the Supplement (Slides 7, 9, 11, and 13). 

Results

- Some results seem unexpected (e.g., the absence of correlation between body composition and heart rate or lactate). Do the authors have explanations or hypotheses for these discrepancies compared to existing literature?

Thank you for the question! The results surprised us as well. The results are likely attributable to the mixed and skill-based components of water polo, which we have elaborated on in detail in the manuscript. We have supplemented the manuscript with the likely reasons for the findings (line 309-315, 320-324).

- The results indicate that height harms some correlations (e.g., VO2abs-max). This effect should be explored further in the discussion, with clear hypotheses explaining how height influences the findings.- Height is mentioned as a confounding factor, but the authors do not sufficiently explain why the effect is negative or how this compares with the existing literature.

Thank you for the comment! In the literature, height has always been reported to positively influence VO2, but this is because body composition/LBM was not taken into account. In our study, by applying a multiple linear regression model, we were able to examine several factors simultaneously, and in our case, LBM had a stronger effect on oxygen uptake than height. However, the significant effect of height improves the accuracy of the model. We have elaborated on this in detail (line 253-259).

- The emphasis on correlations should not use red font but bold text for better clarity.

Thank you for the comment. We have made the corrections.

- The text preceding the tables is highly repetitive and should be simplified, leaving detailed explanations for the discussion section.

Thank you for the feedback, we have simplified them accordingly.

- In the result tables, there is inconsistency in the presentation of results, both in formatting (some are in scientific notation) and in the number of decimal places. Formatting should be consistent and appropriate for the parameters presented.

Thank you for the comment. We have made the corrections.

- Figure captions should be placed below the respective figures.

Thank you for the comment. We have made the corrections.

Conclusion

- The introduction highlights the need to establish normative values for young athletes, but the conclusion does not sufficiently develop how the findings could contribute to this.

- There should be a stronger connection between the objectives presented in the introduction and the conclusions.

- What practical impact do you expect these findings to have on training or evaluating elite athletes?

Thank you for the feedback. With the more precise results derived from body composition, we can define more accurate, sport-specific normative values, which aid in personalized diagnosis. The more accurate assessment would lead to earlier identification of individual deficiencies, helping both the athlete and the coach in designing appropriate training plans. A specific calculator developed for the complex evaluation of body composition and CPET data of youth athletes - possibly by giving percentile values - would facilitate practical use of our results.

We have updated the Conclusion section of the article accordingly (lines 353-366).

- The study limitations should be presented immediately following the discussion.

Thank you for the comment. We have made the corrections.

Reviewer 2 Report

Comments and Suggestions for Authors

Dear authors: Your work seems to be well done but there are some inconsistencies. In order to improve your paper I suggest some modifications. Firstly, the title should include the female gender. Secondly, you try to relate body composition (variable) to CPET (method) which is inappropriate. They should rewrite the text because it is better to relate BC to the variables assessed (VO2abs-max, heart rates, lactate values) or any other variable, and there is a lot of literature that does relate these variables. 

Reference 12 justifying the use of DEXA is too old and obsolete as, in those years, the type of DEXA used was very different from the instrumentation used today - please update this field of knowledge.

On line 60 it is stated that DEXA is the gold standard. This is not the case.A more accurate method is peripheral QCT which provides separate information on trabecular and cortical bone with a low radiation dose, and would therefore be the ideal technique for bone mass studies in children. However, for economic and simplicity reasons, DEXA is widely used.

.From this derives a fundamental issue in its study, which is the use of ionising radiation in adolescents (some of whom are still girls). Although DEXA administers a small amount of ionising radiation, it should not be used in healthy children and adolescents. And its use in sick children would only be restricted to the case where it would provide information necessary for the management of a particular child's disease.

Please indicate whether the informed consent protocol provided to the parents clearly indicated that the girls were going to be exposed to a source of ionising radiation and its possible risks. 

Since the mean age of your participants is 16.10±1.63 y.o. we understand that many of your participants were girls or adolescents (age range: 13.8-19.9). In line 88 you indicate that the participants were <20 y.o. Please clarify this discrepancy. Therefore, results from girls are mixed with results from adolescents and, since the interpretation of DEXA densitometry is very different in children than in adults. 

In fact, I think the results should be separated by age ranges.  I assume that the participating players competed in different categories for age reasons, so presumably the variables studied are also related to age. Therefore, they should display the results according to age groups, e.g. U19, U17, U15. 

Regarding the chronology, given that most of the variables assessed are highly dependent on the state of sporting performance, please clarify whether the measurements were carried out in the same phase of the sporting season or whether they were carried out randomly or by convenience. 

In the results there are no data concerning the variables resting 12-lead electrocardiogram (ECG), blood pressure, measurement, laboratory and tests, echocardiography Please justify this deficit.

Please move lines 146-148 to section 2.1. Study population.

The statement written in lines 93-95 (‘Given that 93 water polo athletes compete...’) assumes that there will be no differences in results in relation to the menstrual cycle although there is consensus that this is not the case, and there are studies that indicate that in the luteal phase (second phase) with higher insulin resistance and higher aldosterone there is usually fluid retention and worsening of sports performance, although much research remains to be done in this field, especially when aerobic variables are evaluated. Please modify or argue your statement. 

Please delete autocite 7 as it is not related to women or water polo. Complete reference 20 What is its relationship to the text (line 90)?

Goodluck to the authors.

Author Response

Dear Reviewer,

Thank you very much for taking the time to review our manuscript. Your insights and comments have greatly contributed to enhancing the professionalism and value of our work. We sincerely appreciate your effort and dedication.

Best regards,
Mark Zamodics 

Firstly, the title should include the female gender.

Thank you for this correction, we agree with the reviewer that emphasizing the female gender of the examined population is important. We have rephrased the title.

Secondly, you try to relate body composition (variable) to CPET (method) which is inappropriate. They should rewrite the text because it is better to relate BC to the variables assessed (VO2abs-max, heart rates, lactate values) or any other variable, and there is a lot of literature that does relate these variables. 

Thank you for this refinement, we have rephrased the text in lines 23 and 94-95 and we have supplemented the Methods section with a new subsection in lines 154-161.

Reference 12 justifying the use of DEXA is too old and obsolete as, in those years, the type of DEXA used was very different from the instrumentation used today - please update this field of knowledge.

Thank you for your feedback, we understand that citing an old and obsolete reference is inadequate in this article. We have replaced reference 12 with a recent review article regarding the use of DEXA in athletes.

On line 60 it is stated that DEXA is the gold standard. This is not the case.A more accurate method is peripheral QCT which provides separate information on trabecular and cortical bone with a low radiation dose, and would therefore be the ideal technique for bone mass studies in children. However, for economic and simplicity reasons, DEXA is widely used.

Thank you for this correction, we agree that we have some more accurate although not easily accessible new methods for bone mass studies in children. We have rephrased the objected sentence in lines 63-64.

… a fundamental issue in its study, which is the use of ionising radiation in adolescents (some of whom are still girls). Although DEXA administers a small amount of ionising radiation, it should not be used in healthy children and adolescents. And its use in sick children would only be restricted to the case where it would provide information necessary for the management of a particular child's disease.

During the DEXA examination, only a small amount of ionising radiation is administered. The radiation exposure is approximately one-thousandth of the natural background radiation that all of us are constantly exposed to during our lifetime. Therefore, the risk of harmful effects is negligible and the study was approved by the Medical Research Council of Hungary (No. IV/10282-1/2020/EKU) following thorough consideration.

Please indicate whether the informed consent protocol provided to the parents clearly indicated that the girls were going to be exposed to a source of ionising radiation and its possible risks. 

The informed consent sheets were sent to the subjects/patients many days before the examinations so they had time to read them carefully as well as they had the opportunity to personally discuss their questions about the examinations with the performing physicians. The sheet regarding the DEXA examination clearly indicated that the children were going to be exposed to a source of ionising radiation and its possible risks. Furthermore, we specifically emphasized that the examination is contraindicated during pregnancy. We have supplemented the Methods section with this information in lines 115-118.

Since the mean age of your participants is 16.10±1.63 y.o. we understand that many of your participants were girls or adolescents (age range: 13.8-19.9). In line 88 you indicate that the participants were <20 y.o. Please clarify this discrepancy. Therefore, results from girls are mixed with results from adolescents and, since the interpretation of DEXA densitometry is very different in children than in adults. 

Thank you for the comment! In our study, we examined the entire Hungarian female national youth team water polo population, which, according to the FINA regulations, includes U16, U18, and U20 teams. Therefore, our study also included athletes over the age of 18. We agree with the Reviewer that bone structure should be evaluated differently for a 15-year-old and a 19-year-old girl, however, our research did not extend to such detailed assessments. In our study, we only considered bone mineral content (BMC) as part of the 3-compartment model.

In fact, I think the results should be separated by age ranges.  I assume that the participating players competed in different categories for age reasons, so presumably the variables studied are also related to age. Therefore, they should display the results according to age groups, e.g. U19, U17, U15. 

Thank you for the feedback! Our study involved members of the U16, U18, and U20 teams (FINA regulations). We also considered analyzing players by specific age groups, but the sample sizes—age<16: 39, age 16-18: 19, age 18-20: 12—would have resulted in statistically unreliable outcomes. Furthermore, many talented young players compete in older age groups or more age groups together, which would also skew the results. For this reason, we applied a multiple linear regression model in our study, which accounts for age – this model is also suitable to demonstrate the changes connected to age. On the other hand, we agree that age groups are important in this population, therefore, we expanded the Methods section to include information on them (line 99). In the future, we plan to expand the examinations and to study age groups of sufficient number also compared to each other. 

Regarding the chronology, given that most of the variables assessed are highly dependent on the state of sporting performance, please clarify whether the measurements were carried out in the same phase of the sporting season or whether they were carried out randomly or by convenience. 

Thank you for this comment, we agree that most of the examined parameters are highly dependent on the state of sporting performance. Fortunately, we had the opportunity to carry out all examinations in the competition season. We have supplemented the Methods section with this information in lines 111-112.

In the results there are no data concerning the variables resting 12-lead electrocardiogram (ECG), blood pressure, measurement, laboratory and tests, echocardiography Please justify this deficit.

The examinations detailed in this article were carried out as part of an extensive cardiology screening of elite athletes. These examinations are mentioned in subsection 2.2 because these helped us to exclude athletes with pathological screening results from further analysis. In other words, these tests also guarantee that only healthy athletes were included in the evaluation. We have supplemented the Methods section with this information in lines 109-110. Unfortunately, the detailed discussion of the results of the cardiology screening of the athletes would go beyond the scope of this communication.

Please move lines 146-148 to section 2.1. Study population.

These information were moved to lines 103-104 of the Study population subsection.

The statement written in lines 93-95 (‘Given that 93 water polo athletes compete...’) assumes that there will be no differences in results in relation to the menstrual cycle although there is consensus that this is not the case, and there are studies that indicate that in the luteal phase (second phase) with higher insulin resistance and higher aldosterone there is usually fluid retention and worsening of sports performance, although much research remains to be done in this field, especially when aerobic variables are evaluated. Please modify or argue your statement. 

Thank you for the comment. This is a very challenging question. The international literature shows conflicting results on this matter, many studies suggest there are no differences in CPET results across the phases of the menstrual cycle, or only minimal differences can be found. Clearly, further targeted research and individual consideration is needed. Accordingly, we have expanded the Limitations section of the article (lines 336-342).

Ref: https://doi.org/10.1080/02701367.2023.2291473

https://doi.org/10.1249/MSS.0000000000003447

https://doi.org/10.1007/s40279-020-01319-3

Please delete autocite 7 as it is not related to women or water polo. Complete reference 20 What is its relationship to the text (line 90)?

Reference 7 as one of our previous publications was cited becasue it holds important information on the specific methods of athlete screening developed by our own laboratory. However we cannot claim that this article is not understandable without these information, therefore we have deleted this reference due to the request of reviewer 2.

Reference 20 was cited as it contains detailed information on the specific features of water polo as well as on the classification of athletes regarding youth groups. We have completed the reference with the missing information.

Round 2

Reviewer 1 Report

Comments and Suggestions for Authors

Dear authors,

I appreciate the efforts taken to revise the article as suggested.

It seems that the indications given were generally accepted and implemented, and the questions were also adequately answered.

Congratulations on your work.

Author Response

Dear Reviewer,

Once again, thank you for your comments and your dedicated work!

Best regards,

Márk Zámodics

Reviewer 2 Report

Comments and Suggestions for Authors

Dear authors, thank you very much for your contributions, although some important suggestions have not been taken into account.

I believe that the document improves in clarity and, if possible, methodological rigour. The title specifies the gender of the participants and the abstract clarifies some important aspects for the understanding of the work.  In the introduction, the paragraph on lines 78-86 better establishes the background of the problem studied.  In the methodology, the characteristics of the participant population have been better identified. 

In the results, the section on correlation analysis between the different variables has been expanded, as well as the statistical analysis.

Unfortunately, no study of the results by age has been carried out, as suggested, which is a serious drawback of both the method and the results, and the regression analysis does not solve this shortcoming. Neither have self-citations 6 and 7 been removed, which, in my opinion, are totally dispensable because there is a large amount of documentation with much more weight in the literature to justify that the CPET examination is an integral part of this extended screening by aiding identification of heart and lung diseases and because in these articles the participants were handball referees who have nothing to do with women water polo players. Nor does the text reflect on the use of ionising radiation in girls and the duty we have to protect at-risk populations such as young women. 

Finally, the conclusions have apparently been rewritten with more accuracy with respect to the results.

Author Response

Dear Reviewer,

Thank you for your feedback!

We have addressed your comments as follows:

The highlighted self-citations have been removed and replaced (line 453-458, ref-6-7.).

We have elaborated on the ionizing radiation associated with DEXA and included reference from expert reports (lines 140-148, 499 – Ref. 23).

Age-group-specific statistics were conducted and incorporated into the Methods, Results, Discussion, and Supplement sections (lines 104-105, 238-246,352-365, Supplement slide 8-9, 12-13,16-17, 20-21).

It is important to emphasize that the potential age-group differences may not be fully reflected in the statistical analysis due to the limited sample size and the numerical distribution of participants across age groups.

Thank you again for your valuable comments and work!

Best regards,

Márk Zámodics!

Round 3

Reviewer 2 Report

Comments and Suggestions for Authors

I believe that the modifications have improved the text of the manuscript.

Best of luck to the authors.